# Molecular profiling of 888 pediatric tumors informs future precision trials and data-sharing initiatives in pediatric cancer

Suzanne J. Forrest [1,2] ✉, Hersh Gupta[3,4], Abigail Ward[1], Yvonne Y. Li [3,4], Duong Doan[1], Alyaa Al-Ibraheemi[2,5], Sanda Alexandrescu[2,5], Pratiti Bandopadhayay [1,2], Suzanne Shusterman[1,2], Elizabeth A. Mullen [1,2], Natalie B. Collins [1,2], Susan N. Chi[1,2], Karen D. Wright[1,2], Priti Kumari[3], Tali Mazor [3], Keith L. Ligon [2,3,5,6], Priyanka Shivdasani[6], Monica Manam[5], Laura E. MacConaill[6], Evelina Ceca[1], Sidney N. Benich[1], Wendy B. London [1], Richard L. Schilsky [7], Suanna S. Bruinooge [7], Jaime M. Guidry Auvil[8], Ethan Cerami[3], Barrett J. Rollins[2,3,6], Matthew L. Meyerson [2,3,4], Neal I. Lindeman[9], Bruce E. Johnson[2,3,6], Andrew D. Cherniack [3,4], Alanna J. Church [2,5] & Katherine A. Janeway [1,2] ✉

To inform clinical trial design and real-world precision pediatric oncology practice, we classified diagnoses, assessed the landscape of mutations, and identified genomic variants matching trials in a large unselected institutional cohort of solid tumors patients sequenced at Dana-Farber / Boston Children's Cancer and Blood Disorders Center. Tumors were sequenced with OncoPanel, a targeted next-generation DNA sequencing panel. Diagnoses were classified according to the International Classification of Diseases for Oncology (ICD-O-3.2). Over 6.5 years, 888 pediatric cancer patients with 95 distinct diagnoses had successful tumor sequencing. Overall, 33% (n = 289/888) of patients had at least 1 variant matching a precision oncology trial protocol, and 14% (41/289) were treated with molecularly targeted therapy. This study highlights opportunities to use genomic data from hospital-based sequencing performed either for research or clinical care to inform ongoing and future precision oncology clinical trials. Furthermore, the study results emphasize the importance of data sharing to define the genomic landscape and targeted treatment opportunities for the large group of rare pediatric cancers we encounter in clinical practice.

Over the last 50 years, there has been a profound prolongation in the survival of children and adolescents with cancer, primarily achieved through the intensification of chemotherapy, risk stratification, and multi-modal treatments[1,2]. Despite these advances, cancer remains the leading cause of death by disease among children in the United States,

and many survivors of childhood cancers have significant long-term sequelae from their treatment[3–6]. Furthermore, progress has not been universal, and a number of specific cancer diagnoses have seen little improvement in outcomes and continue to have a disproportionate burden of treatment-related side effects. Specific groups with lagging

[1]Dana-Farber/Boston Children's Cancer and Blood Disorders Center, Boston, MA, USA. [2]Harvard Medical School, Boston, MA, USA. [3]Dana-Farber Cancer Institute, Boston, MA, USA. [4]Broad Institute of MIT and Harvard, Cambridge, MA, USA. [5]Boston Children's Hospital, Boston, MA, USA. [6]Brigham and Women's Hospital, Boston, MA, USA. [7]American Society of Clinical Oncology, Alexandria, VA, USA. [8]National Cancer Institute, Bethesda, MD, USA. [9]Weill Cornell Medical College, New York, NY, USA. ✉e-mail: Suzanne_Forrest@dfci.harvard.edu; Katherine_Janeway@dfci.harvard.edu

**Fig. 1 | CONSORT diagram of analytic cohort.** Flow chart illustrating derivation of the final study cohort.

improvement in outcomes include pediatric, adolescent, and young adult (AYA) brain tumors and sarcomas[1].

One of the major drivers of improved outcomes and decreased toxicity of therapy is the application of precision oncology, carried out by cancer type-specific characterization of the genome paired with molecularly targeted therapy using investigational or approved agents[7,8]. In many cancers, genomic characterization has also facilitated the classification of biological subtypes associated with treatment response and resistance and the development of risk-stratified treatment protocols[8–10]. Pediatric brain and extracranial solid tumors are a group of ultra-rare malignancies occurring in pediatric and AYA patients[11]. The rarity of many pediatric brain and solid tumors is a barrier to generating clinical-genomic databases containing sufficient patients for meaningful genomic analyses to guide precision oncology and has slowed progress in these diseases.

At Dana-Farber / Boston Children's Cancer and Blood Disorders Center, since 2013, all pediatric patients with cancer or suspected cancer have been eligible for enrollment in the Profile Cancer Research Study, an institutional sequencing study generating clinical grade targeted next-generation sequencing (NGS) reports which are returned to treating physicians and the medical record[12]. The study was universally offered to pediatric brain and solid tumor patients. The resulting data presents an opportunity to perform analyses of the genomic features of these rare and ultra-rare pediatric cancers, facilitating clinical trial design and real-world practice in precision oncology. In addition, these data can be contributed to data-sharing initiatives in pediatric cancer, including the National Cancer Institute's (NCI) Childhood Cancer Data Initiative (CCDI)[13,14].

## Results

### Patient and sample characteristics: sequenced cases have a long tail of ultra-rare diagnoses

Between September 2013 and March 2019, 1120 pediatric patients with intracranial (CNS) or extracranial solid tumors consented to and enrolled in the Profile Cancer Research Study. Targeted NGS of tumor samples was performed on tissue obtained at the time of a clinically indicated procedure. The OncoPanel assay, performed at the Center for Advanced Molecular Diagnostics (CAMD) at the Brigham and Women's Hospital, was successful for 76% (848/1120) of enrolled

participants. Sufficient tumor tissue from a previous clinical procedure was unavailable for 201 of the 1120 enrolled participants (18%). Seventy-one participants (6%) had insufficient or low-quality tumor DNA after extraction (Fig. 1).

OncoPanel data from an additional 27 patients enrolled at Dana-Farber in a similar, previously published study[15], were also included in this analysis. Data from 13 patients with extracranial solid tumors sequenced with OncoPanel with a waiver of informed consent were also included in the analysis (Fig. 1).

The final analytic cohort included 888 pediatric patients with solid tumors with successful somatic OncoPanel sequencing. Within this analytic population, 512 (58%) had extracranial solid tumors and 376 (42%) had CNS tumors. Median age at cancer diagnosis was 7.66 years; 56% of the patients were male, and 65% were white. The tumor stage at diagnosis was localized for 60% of the extracranial solid tumors, while 29% had metastatic disease. Most samples, 92%, were from the primary tumor site, and 73% were obtained at initial diagnosis prior to treatment. Most of the samples were sequenced with OncoPanel versions 2 and 3, with 12.6% sequenced with version 1 (Table 1).

Patient diagnoses were confirmed and uniformly classified according to ICD-O-3.2 by a multidisciplinary team review of the pathology reports. 95 distinct histologic cancer diagnoses were represented in the study cohort. While 55% (451/888) of the patients in the cohort had one of ten common pediatric cancers, the remaining 45% of participants (398/888) had one of 85 distinct rare pediatric cancer diagnoses. The common pediatric cancer diagnoses were: neuroblastoma ($n = 78$), low-grade astrocytoma ($n = 72$), Wilms tumor ($n = 58$), medulloblastoma ($n = 55$), pilocytic astrocytoma ($n = 47$), rhabdomyosarcoma ($n = 44$), osteosarcoma ($n = 42$), ependymoma ($n = 39$), Ewing sarcoma ($n = 28$), and glioblastoma multiforme ($n = 27$) (Fig. 2). For the 85 distinct rare pediatric cancers, there were fewer than 25 patients per histologic diagnosis.

### Sequenced cases are representative of national pediatric cancer registries and contain diagnoses not present in prior pan-pediatric cancer sequencing analyses

Proportionally, extracranial solid tumor diagnoses in this analytic cohort are similar to the National Cancer Institute (NCI) National Childhood Cancer Registry (NCCR) pediatric (ages < 20) population

## Table 1 | Clinical Characteristics of Patients and Sequenced Samples

| | Solid Tumors $N = 512$ | CNS Tumors $N = 376$ | All $N = 888$ |
|---|---|---|---|
| **Age at diagnosis (years), n (%)** | | | |
| 0–5 | 220 (43.0%) | 156 (41.5%) | 376 (42.3%) |
| 6–10 | 81 (15.8%) | 93 (24.7%) | 174 (19.6%) |
| 11–15 | 113 (22.1%) | 74 (19.7%) | 187 (21.1%) |
| 16–20 | 66 (12.9%) | 44 (11.7%) | 110 (12.4%) |
| >20 | 18 (3.5%) | 3 (0.8%) | 21 (2.4%) |
| Unknown | 14 (2.7%) | 6 (1.6%) | 20 (2.3%) |
| **Sex, n (%)** | | | |
| Male | 288 (56.2%) | 213 (56.6%) | 501 (56.4%) |
| Female | 224 (43.8%) | 163 (43.4%) | 387 (43.6%) |
| **Race, n (%)** | | | |
| White | 331 (64.6%) | 246 (65.4%) | 577 (65.0%) |
| Black or African American | 17 (3.3%) | 10 (2.7%) | 27 (3.0%) |
| Asian | 23 (4.5%) | 22 (5.9%) | 45 (5.1%) |
| Native Hawaiian or Other Islander | 1 (0.2%) | 0 (0%) | 1 (0.1%) |
| Some other race | 45 (8.8%) | 40 (10.6%) | 85 (9.6%) |
| Multiple Race | 4 (0.8%) | 1 (0.3%) | 5 (0.6%) |
| Unknown/Not specified | 91 (17.8%) | 57 (15.2%) | 148 (16.7%) |
| **Ethnicity, n (%)** | | | |
| Hispanic or Latino | 41 (8.0%) | 17 (4.5%) | 58 (6.5%) |
| Non-Hispanic or Latino | 452 (88.3%) | 358 (95.2%) | 810 (91.2%) |
| Unknown | 19 (3.7%) | 1 (0.3%) | 20 (2.3%) |
| **Stage at diagnosis, n (%)** | | | |
| Localized | 307 (60.0%) | 119 (31.6%) | 426 (48.0%) |
| Regional | 25 (4.9%) | 0 (0%) | 25 (2.8%) |
| Metastatic | 149 (29.1%) | 18 (4.8%) | 167 (18.8%) |
| Unknown | 18 (3.5%) | 1 (0.3%) | 19 (2.1%) |
| Not applicable | 13 (2.5%) | 238 (63.3%) | 251 (28.3%) |
| **Timing of sequenced sample, n (%)** | | | |
| Initial diagnosis before treatment | 365 (71.3%) | 283 (75.3%) | 648 (73.0%) |
| Local control at initial diagnosis | 80 (15.7%) | 15 (4.0%) | 95 (10.7%) |
| Relapse/progression before treatment | 38 (7.4%) | 57 (15.2%) | 95 (10.7%) |
| Relapse/progression local control | 13 (2.5%) | 11 (3.0%) | 24 (2.7%) |
| Autopsy | 1 (0.2%) | 0 (0%) | 1 (0.1%) |
| Unknown | 15 (2.9%) | 10 (2.7%) | 25 (2.8%) |
| **Biopsy/resection site, n (%)** | | | |
| Primary tumor | 443 (86.5%) | 369 (98.1%) | 812 (91.5%) |
| Metastatic site | 66 (12.9%) | 1 (0.3%) | 67 (7.5%) |
| Unknown | 3 (0.6%) | 6 (1.6%) | 9 (1.0%) |
| **OncoPanel version, n (%)** | | | |
| Version 1 | 68 (13.3%) | 34 (9.4%) | 112 (12.6%) |
| Version 2 | 221 (43.2%) | 175 (46.5%) | 396 (44.6%) |
| Version 3 | 223 (43.6%) | 167 (44.4%) | 390 (43.9%) |

from 2014–2018[11], with the exception of carcinomas which were underrepresented in our cohort (6.8%) compared to NCCR cohort (26.8%) (Fig. 3). This difference is likely driven by adolescent and young adult patients in the NCCR registry with thyroid carcinomas (14.5%), as these patients are often not referred to our pediatric oncology practice. There are two often cited landmark pediatric pan-cancer sequencing

analyses[16,17]. All 10 common pediatric diagnoses are included in these pan-pediatric cancer analyses, while 75 of the 85 (88%) rare cancer diagnoses in our analytic population are not represented (Fig. 2).

### Pediatric and AYA cancers frequently harbor genomic variants matching targeted therapy basket trials but real-world precision oncology practice often involves off-label treatment

To identify patients with tumor variants that would have constituted eligibility criteria for a clinical trial of a molecularly targeted therapy or for which there are clinically indicated agents, we used the actionable mutation of interest (aMOI) lists from three large precision medicine basket trials: the NCI-Children's Oncology Group (COG) Pediatric MATCH (Molecular Analysis for Therapy Choice) trial[18]; the NCI-MATCH trial[19]; and the ASCO (American Society of Clinical Oncology) TAPUR (Targeted Agent and Profiling Utilization Registry) Study (Supplementary Data 1)[20]. Overall, 33% ($n = 289/888$) of patients had tumors with at least 1 oncogenic genomic alteration that matched to a targeted treatment arm of at least one of the three precision oncology basket trials (Fig. 4)[18–20]. The number of patients with tumors that had an actionable variant matching a treatment arm of the NCI-COG Pediatric MATCH Trial, the NCI-MATCH Trial, and the ASCO TAPUR Study were 238, 227, and 124, respectively. Seventy-five patients had a tumor variant matching an arm of all 3 trials. The genes that most frequently contained aMOIs were: BRAF (10%), NF1 (4%), CDKN2A (4%), PI3KCA (2.4%), NRAS/KRAS (2.1%), BRCA2 (1.5%), ALK (1.2%), and FGFR1 (1.2%) (Fig. 4).

Many patients ($n = 219$) had tumors with variants that matched multiple treatment arms, and some patients ($n = 64$) had tumors with multiple matching genomic alterations. The proportion of patients with variants matching precision oncology treatment protocols differed by diagnosis. Glioneuronal tumors, high-grade gliomas, and pilocytic astrocytomas had the highest match rates at 89%, 70%, and 64%, respectively, driven by BRAF alterations (Supplementary Table 1). Ewing sarcoma and Wilms tumor had the lowest match rate with only 7% and 12%, respectively, of patients with those diagnoses having a tumor variant matching a targeted therapy treatment arm (Supplementary Table 1).

We reviewed the prescription data and medical records of patients with tumors containing precision medicine basket trial aMOIs to determine whether they had received molecularly targeted therapy matching the identified aMOI. Of the 289 patients who had tumors with aMOIs, 41 (14%) received 48 matched molecularly targeted therapies (Fig. 5). Two patients received 2 matched therapies, and 2 patients received 3 matched therapies. Ten classes of molecularly targeted therapies were received, as shown in Fig. 5. Only 12% (5/41) of the patients who received a matched molecularly targeted therapy were enrolled in a clinical trial, with 88% (36/41) of patients receiving the matched therapy via single patient protocol or off-label (Fig. 5).

### Analysis of recurrent genomic alterations in rare, previously understudied pediatric cancers reveals unique opportunities for targeted therapy

We analyzed recurrent genomic alterations in this patient population separately for extracranial solid malignancies and CNS tumors. We then also assessed the oncogenic mutations uniquely enriched in the 75 rare extracranial solid tumors and CNS diagnoses that were not included in the two previously reported large pan-pediatric cancer analyses[16,17]. The most common recurrent oncogenic alterations occurring in >5% of the extracranial solid tumor cases were TP53 mutation/deletions (9%), MYC/MYCN amplifications (6%), and CTNNB1 mutations (6%) (Supplementary Fig. 1). In the CNS tumors, the most common oncogenic alterations occurring in >5% of the cases were BRAF rearrangements/mutations (21%), TP53 mutations/deletions (12%) and CDKN2A/B deletions (6%) (Supplementary Fig. 2).

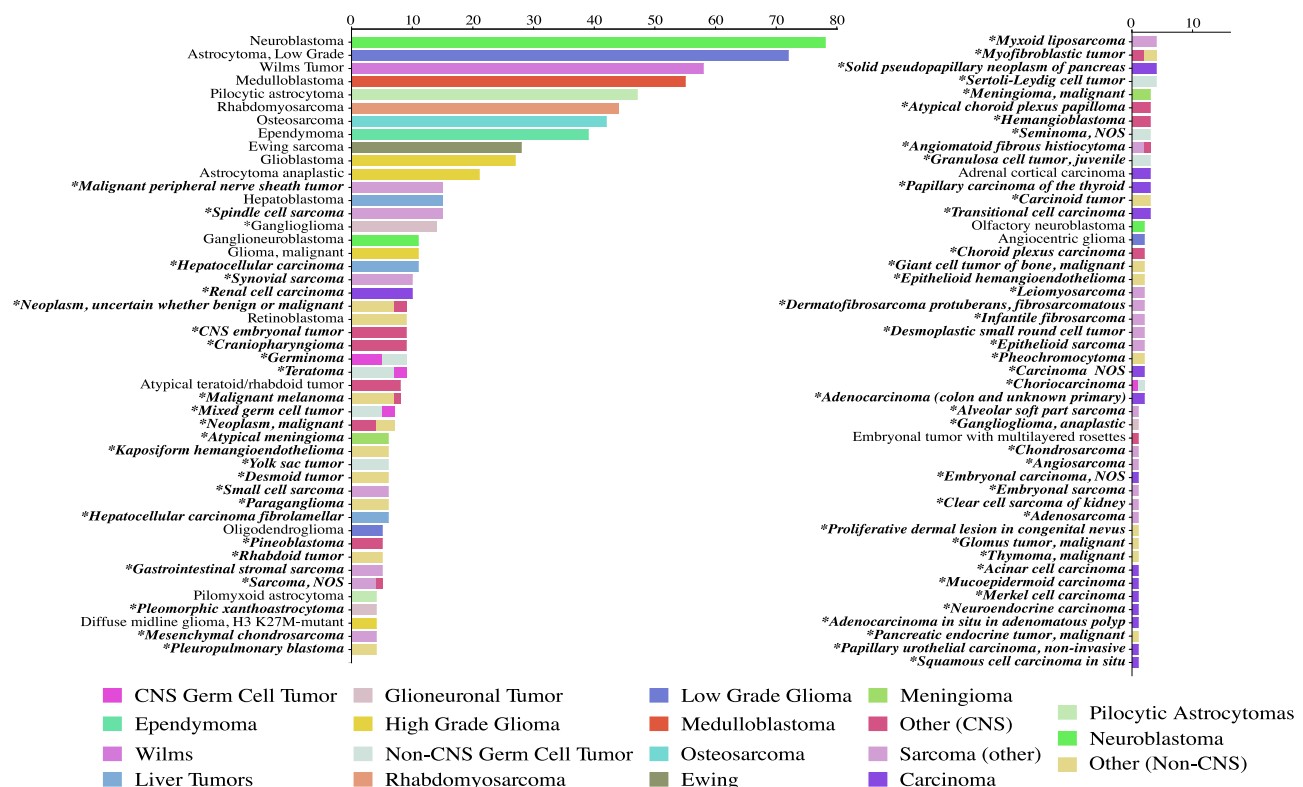

**Fig. 2 | Longtail of patient diagnoses classified by ICD-O-3.2.** The number of tumors sequenced with each pathologic diagnosis are shown. Diagnoses are color coded by disease sub-group. Diagnoses marked with * and in bold were not included in two prior pediatric pan-cancer sequencing studies[16,17].

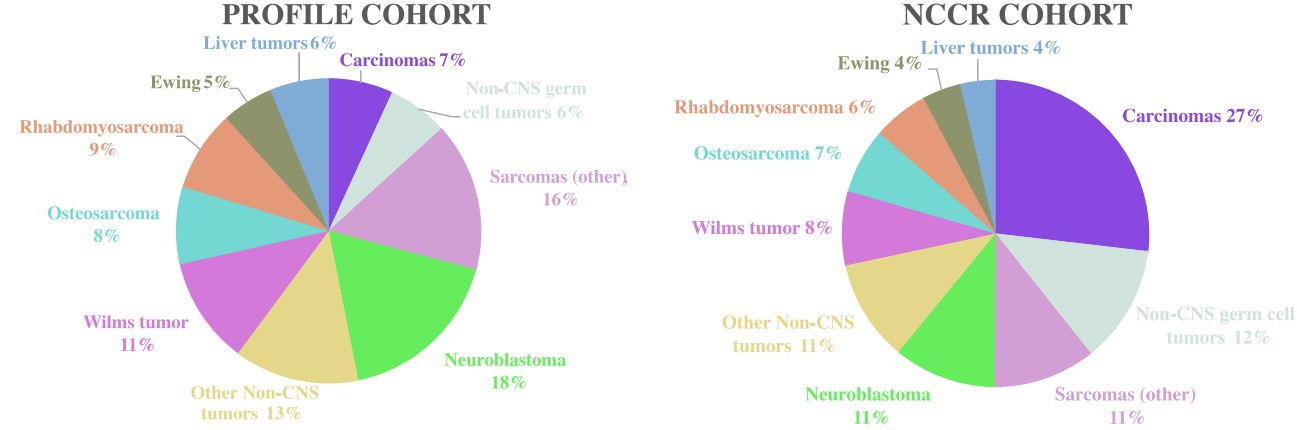

**Fig. 3 | The distribution of pediatric extracranial solid tumor diagnoses represented in this study cohort compared to the NCCR Registries 2014-2018.** The percentage of patients with each extracranial solid tumor diagnosis or diagnosis group is shown.

We observed the expected patterns of genomic events in specific diagnoses, including *TP53* rearrangements in osteosarcoma, *EWSR1* rearrangements in Ewing sarcoma, *ALK* mutations and *MYCN* amplification in neuroblastoma, *BRAF* fusions and *IDH1* mutations in low-grade glioma and *TP53* mutations, *CDKN2A/B* deletions and *H3F3A* mutations in high-grade gliomas. In contrast, activating *PIK3CA* gene alterations, present in 18 cases (2% of the entire cohort), were distributed across cancer diagnoses with 6 different extracranial solid tumors and 5 different CNS solid tumor diagnoses containing these alterations. Similarly, *ARID1A* inactivating mutations, present in 10 cases (1.1% of the entire cohort), were present in 8 different histologies (8 extracranial and 2 CNS) (Supplementary Table 2, Supplementary Fig. 1 and Supplementary Fig. 2).

In the 235 extracranial solid tumors with histologies not included in prior pediatric pan-cancer analyses, the recurrently altered genes uniquely containing oncogenic alterations (compared to the common tumors) were *CTNNB1*, *DICER1*, and *NF1* (Fig. 6a). In the 78 CNS tumors with histologies not included in prior pediatric pan-cancer analyses, the recurrently altered genes uniquely containing oncogenic alterations (compared to the common tumors) were *CTNNB1*, *NF2* and *KIT* (Fig. 6b). Genomic alterations potentially targetable with precision therapeutics uniquely present in the rare CNS and extracranial solid tumors include *ERBB2* activating mutations in carcinomas, *KIT* activating mutations in CNS and non-CNS germ cell tumors, and *CTNNB1* inactivating mutations in carcinomas, liver tumors, desmoid tumors, and craniopharyngiomas (Fig. 6). In addition to these therapeutically

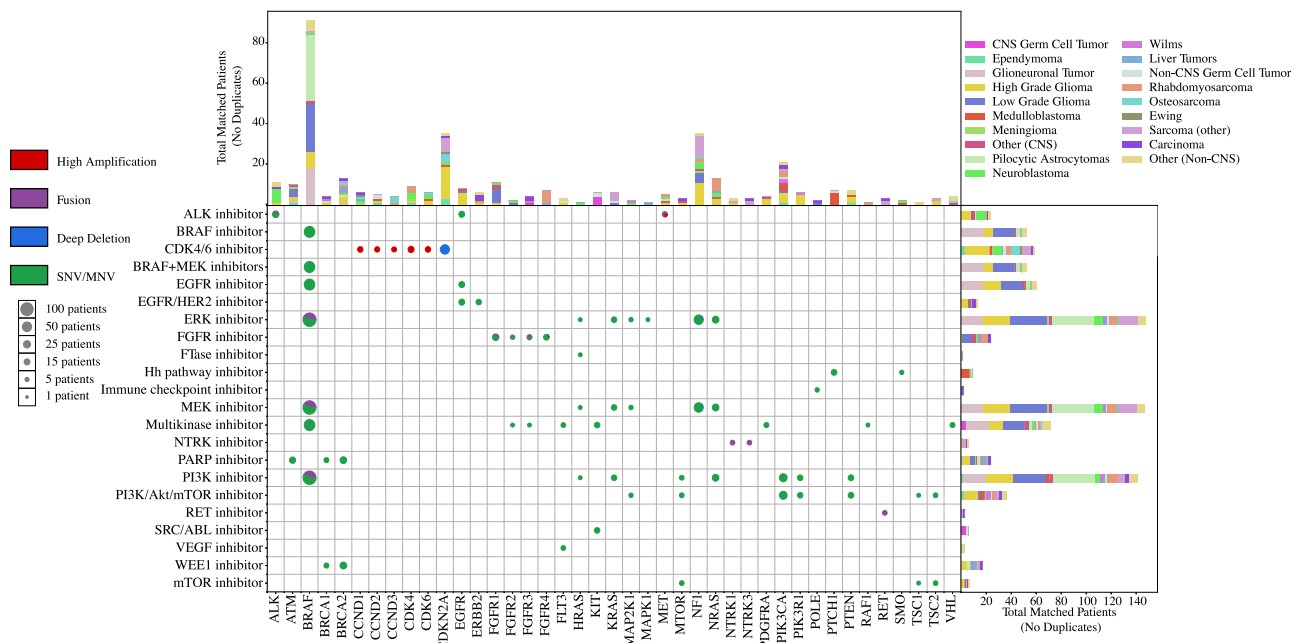

**Fig. 4 | Patients with oncogenic tumor variants matching to a treatment arm of the NCI-COG Pediatric MATCH Trial, NCI-MATCH Trial, or ASCO TAPUR Study.** Gene altered and corresponding matching targeted therapy type are shown. The size of each dot represents the number of patients with a matching variant in that gene, and the color of the dot represents the type of variant. Matched patients' diagnosis grouping is shown by gene and type of targeted treatment.

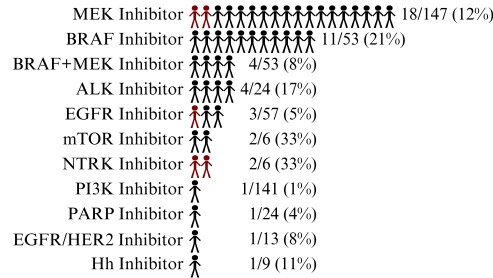

**Fig. 5 | Patients who received molecularly targeted therapy to identified aMOIs.** The number and proportion of patients who received matched therapy is shown by inhibitor type. The black color indicates receipt of matched therapy via single patient protocol or off-label, and the red color indicates receipt of matched therapy on a clinical trial.

actionable variants, *DICER1* alterations were present in 4% (13/313) of the rare diagnoses, with potential implications for both diagnosis and germline cancer predisposition (Fig. 6).

**Sequencing data linked to electronic medical records presents an opportunity for data sharing**

The sequencing data for this study were shared with the NCI's CCDI. Because these sequencing data retained a link to the patient's Dana-Farber / Boston Children's Cancer and Blood Disorders electronic medical record (EMR), there was the opportunity to annotate these genomic data more fully to provide a complete picture of the cancer diagnosis, treatment, and outcome. For selected osteosarcoma, Ewing sarcoma, Wilms tumor, and neuroblastoma patients included in the analytic population, we used the PRISSMM™ data model[21] and trained data abstractors to obtain these additional clinical data from the EMR. The EMR for 38, 29, 25, and 20 patients with Wilms tumor, osteosarcoma, Ewing sarcoma, and neuroblastoma, respectively, were abstracted into the PRISSMM™ data model in RedCap (Research Electronic Data Capture)[22,23]. For these patients, the deeper clinical annotation includes Toronto Stage[24] on 92 patients, an average of 4 pathology reports, and 21 imaging reports per patient, with a range of

1–29 reports per patient for pathology and 1–110 reports per patient for imaging. The average number of cancer treatment regimens captured in the data per patient is 3, with a range of 1–12. The median follow-up time for these patient data is 38 months with a range of 0-300 months.

## Discussion

Patients with CNS and extracranial solid tumors seen by pediatric oncology at Dana-Farber/Boston Children's Cancer and Blood Disorders Center were offered the opportunity to participate in a cohort study of tumor profiling with the return of results. The study enabled us to generate, analyze, and share sequencing and clinical data for both common and rare pediatric malignancies. Initial efforts sequencing pediatric cancers, such as the Therapeutically Applicable Research to Generate Effective Treatments (TARGET) Childhood Cancer Program, understandably focused on more common pediatric cancers[25–28]. As a result, the initial pan-pediatric cancer mutational landscape analyses were biased towards more common pediatric solid tumors and cancer types of specific interest to the investigators. In contrast, studies like ours enrolling any pediatric cancer patient in a cohort study with the return of sequencing results generate data on a complete spectrum of pediatric solid tumors. As shown in Table 2, Genomes for Kids and Pediatric MATCH enrolled a similarly large number of distinct diagnoses[29,30]. Due to the prevalence of diagnoses in the long tail of diverse pediatric solid tumors, biologic insights for the ultra-rare cancers in this and similar studies will require data sharing to fully exploit the potential therapeutic opportunities. For example, a combined dataset would permit a better understanding of genomic data presented in this analysis, such as a more precise estimate of the frequency of targetable *PIK3CA* mutations in pediatric solid tumors and a better understanding of the diagnoses in which they occur. We have contributed the data from a subset of this analytic cohort to GENIE (Genomics Evidence Neoplasia Information Exchange) and from the entire cohort to the NCI's Childhood Cancer Data Initiative (see "Methods", Data Availability sections for more information).

Shared genomic data with paired clinical data, including baseline disease features, treatment, and outcomes, will allow the scientific

**a**

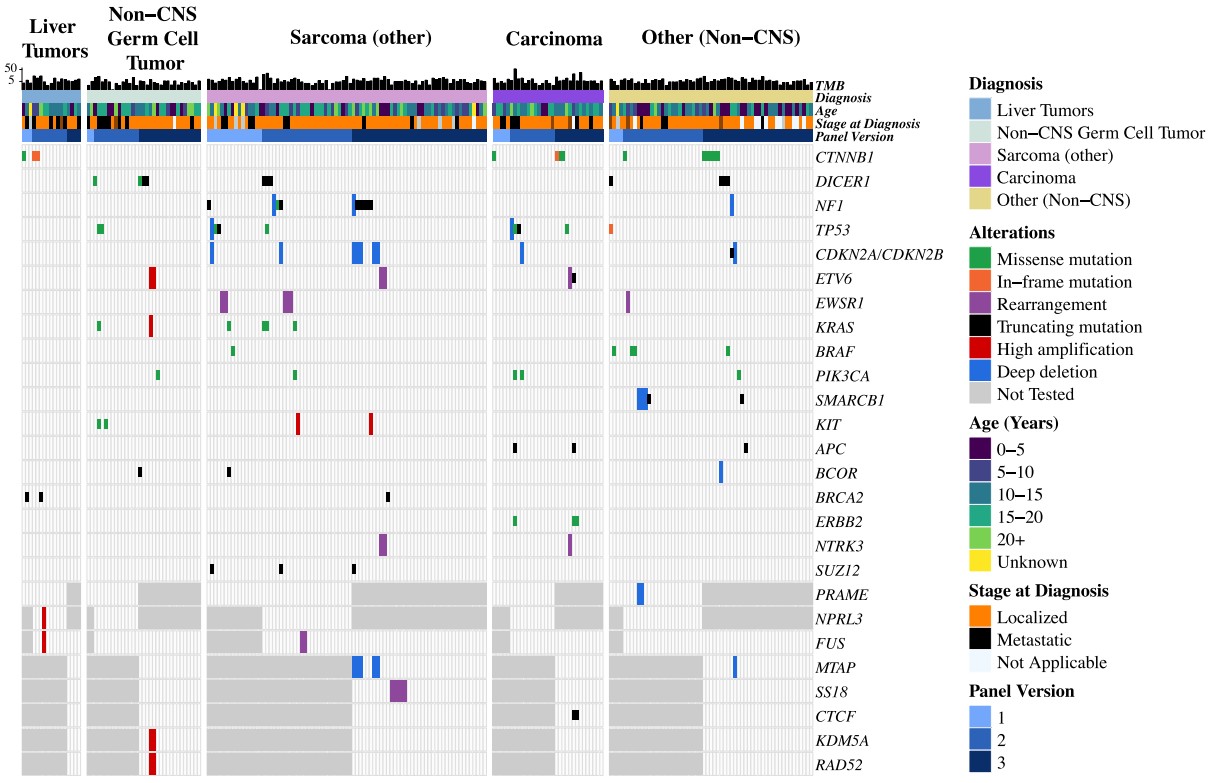

**b**

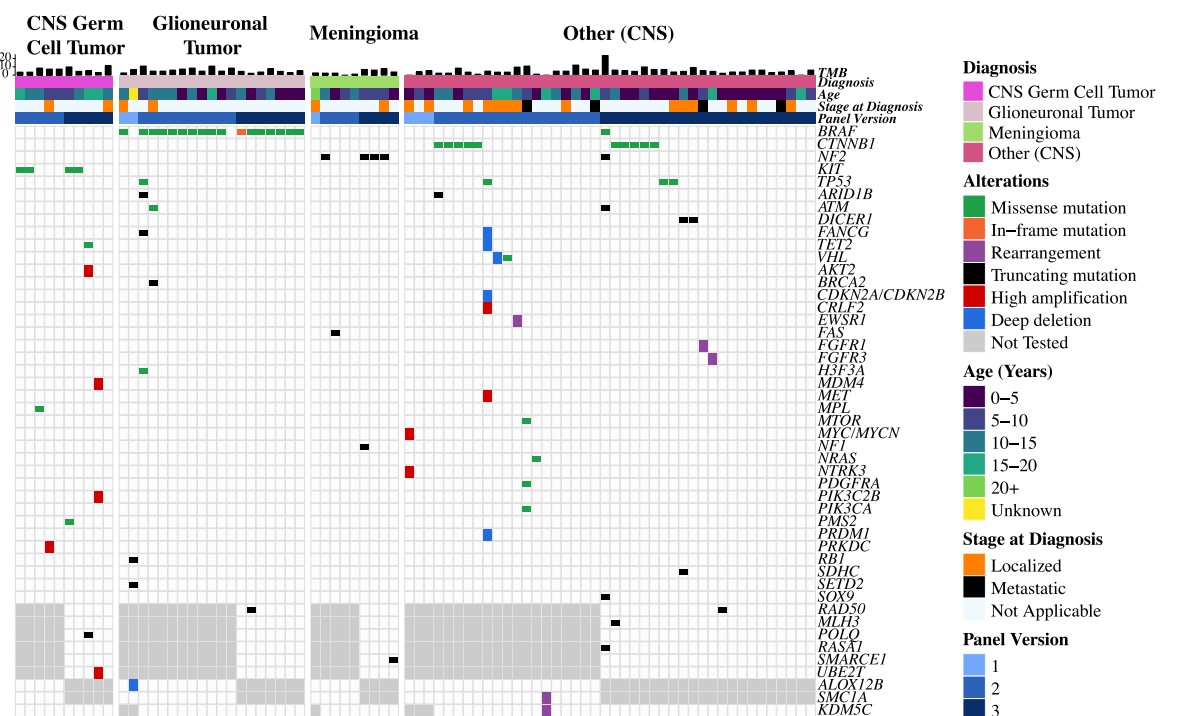

**Fig. 6 | OncoPrint showing most common oncogenic alterations in the histologies not included in two prior pediatric pan-cancer sequencing studies.** Alterations with >1% frequency within the shown patients are displayed along with clinical features of each case for **a** Extracranial solid tumors (*n* = 235) and **b** CNS tumors (*n* = 75).

community to identify genomic features of disease that are biomarkers for poor outcomes or that predict treatment response or resistance mechanisms. As such, developing standards for categorizing and reporting clinical data from the EMR is an ongoing high priority. Given the number of rare pediatric solid tumors, diagnosis classification is an essential component of data standardization. Here, we show that classifying pediatric cancer diagnoses using the ICD-O standard ontology system is feasible, and we believe a prospective approach to diagnosis classification within the EMR would benefit research use of clinical data and help facilitate current and future data-sharing

**Table 2 | Characteristics of current study and 5 recently published clinical sequencing/matching protocols**

|  | Current | Genomes for Kids | INFORM | MAPPYACTS | Pediatric-MATCH | Zero Childhood Cancer |
|---|---|---|---|---|---|---|
| Patients, n | 888 | 253 | 926 | 624 | 1000 | 247 |
| Age, y | 0–25 | 0–25 | 0–40 | 0–38 | 1–21 | 0–31 |
| Diagnosis Group, % |  |  |  |  |  |  |
| CNS | 42% | 31% | 27% | 29% | 25% | 37% |
| HM | – | 41% | 9% | 10% | 3% | 17% |
| ST | 58% | 27% | 64% | 61% | 72% | 46% |
| Disease Status, % |  |  |  |  |  |  |
| Initial diagnosis | 84% | 85% | – | – | – | 47% |
| R/R or high-risk | 14% | 15% | 100% | 100% | 100% | 51% |
| Histologies, n | 95 | 81 | 17 + other | 36 | 80 | 24 + other |
| Type of Sequencing | Panel | WES | WES | WES | Panel | WGS |
|  |  | WGS | icWGS | RNAseq |  | RNAseq |
|  |  | RNAseq | RNA seq Methylation | Panel |  | Methylation |
| Year Published |  | 2021 | 2021 | 2022 | 2022 | 2020 |
| PMID |  | 34301788[29] | 34373263[37] | 35292802[32] | 35353553[30] | 33020650[55] |

initiatives in pediatric solid tumors. In addition, we adapted the PRISSMM™ model to several pediatric cancers and shared this data, with a code enabling linkage to genomic data, with the CCDI's NCCR initiative.

Molecular tumor profiling tests are becoming a more common component of clinical care for pediatric patients with solid malignancies[15]. Observations from this study shed light on what clinicians can expect when sequencing pediatric patients with solid malignancies in the clinic. Similar to other studies, almost 20% of the patients enrolled in this study did not have tumor tissue available (8% had no procedure, 9% had insufficient tumor) for sequencing[31,32]. This is partly due to a high rate of second opinion visits in our pediatric brain and solid tumor programs. Other contributing factors include small biopsies for which leftover specimens were not available and past tissue processing practices in pathology, such as harsh decalcification of bone sarcomas[33]. The technical failure rate of 6% is consistent with failure rates previously reported by us and other groups, including the Pediatric MATCH trial[30]. We report a 33% rate of detecting genomic variants which would make patients potentially eligible for clinical trial arms of basket precision oncology trials, which is essentially the same as the rate reported (31.5%) for the Pediatric MATCH screening protocol[30]. Of note, the proportion of patients matching precision oncology treatment trials was the same for the Pediatric and Adult MATCH trials supporting current drug development policies of considering pediatric trials for molecularly targeted therapies[34,35].

These results highlight the importance of obtaining molecular characterization of pediatric CNS and extracranial solid tumors. The extent to which molecular testing has been incorporated into the standard care of these patients varies by treatment setting, country, and diagnosis. For example, there are national research programs in the United States and several European countries offering sequencing, including, in many cases RNAseq, for either all pediatric cancers or specific diagnoses[36–38]. In the United States, there are national guidelines for NGS of adult cancers, which provides support for insurance reimbursement for molecular testing[39–41]. However, guidelines for pediatric tumors are more limited but beginning to be established for select diagnoses[42,43]. Continued efforts to address the role of molecular profiling for pediatric cancers will be important as these guidelines are developed. Diagnosis-specific guidelines will not fully address the role of sequencing in the ultra-rare histologies constituting 45% of the cases in this study, and an ultra-rare pediatric cancer or pan-pediatric cancer guideline or statement will likely be needed. In the interim, continued efforts to molecularly profile pediatric CNS and

extracranial solid tumors, particularly ultra-rare and advanced cancers, for diagnostic, prognostic, and therapeutic purposes are critically important.

Only 14% of the 289 patients with an aMOI and 5% of the overall sequenced population received molecularly targeted therapy matched to an identified genomic variant. There are several possible reasons for the low proportion of patients with an aMOI treated with targeted therapy, which could be explored in future studies. Patients were most often enrolled at diagnosis and may have completed standard therapy without the need for further therapy. Furthermore, treatment data may be missing for a subset of patients seen for second opinions. Lastly, the era in which the sequencing was performed extends back to 2013, when fewer molecularly targeted treatments were available. Interestingly, molecularly targeted therapy was received outside of a clinical trial for 88% of patients, a similar finding to our recent report in relapsed and refractory extracranial solid tumors (72% of patients treated off trial)[44]. This finding suggests the need for future efforts to collect high-quality treatment and outcome data from EMRs in order to understand dosing, administration, efficacy, and toxicity data for molecularly targeted therapies used in pediatric solid tumor patients.

The major limitation of this study is that sequencing was from tumor only and utilized a targeted panel. As a result, it is challenging to perform mutational signature analysis. Furthermore, sequencing data may, in some cases, contain germline variants that could be inappropriately classified as somatic[45]. At Dana-Farber Cancer Institute, in collaboration with the Brigham and Women's Hospital, we have now launched a paired tumor-germline targeted sequencing assay to address these issues and patients are currently eligible to enroll in a study to access this assay.

## Methods
### Study participants and tumor samples
This study complies with all relevant ethical regulations and was approved by the Dana-Farber Cancer Institute (DFCI) Institutional Review Board (IRB). All patients seen at Dana-Farber/Boston Children's Hospital Cancer and Blood Disorders Center with a suspected or confirmed cancer were eligible to participate in the Profile Cancer Research Study starting in 2013[12]. There was no age limit, and patients were considered pediatric if they were seen by a pediatric provider. Between 2013 and 2019 all pediatric patients with a brain tumor or extracranial solid tumor were offered the opportunity to enroll in the study. Patients and their families who provided written informed consent underwent targeted NGS sequencing of tumor specimens collected for clinical purposes (eg. biopsy or resection) in a Clinical

Laboratory Improvement Amendments (CLIA)-certified clinical laboratory. Sequencing results were returned to the physician and medical record. Pediatric patients consented to the Profile Study between September 2013 and March 2019 with a solid tumor and successful tumor sequencing were included in this analysis. More than four years have passed since the last patient was enrolled, so adequate time has now passed for many of the patients to utilize the genomic information for potential treatment based on the genomic findings. Pediatric patients enrolled on Profile with a hematologic malignancy, or a benign tumor were excluded. Tumor samples were requested from the pathology department following patient consent after the standard pathology evaluation was complete. Tumor sample acquisition procedures were not altered for these studies, and these clinical samples were most often leftover FFPE (formalin fixed paraffin embedded) specimens in the pathology department. Several additional patients who underwent targeted NGS sequencing with Onco-Panel on a similar multi-institution sequencing study[15] and a small number of tumor samples sequenced under a waiver of consent were also included if they had a spindle or round cell sarcoma. If patients had multiple tumor samples sequenced at different time points, then only one was used for analysis. Tumor sample acquired at the time of initial diagnosis was used when available, and if not available, the sample with better quality was used based on pre-determined criteria (pre-treatment samples at the earliest available relapse/recurrence were prioritized over post-treatment specimens).

### Clinical data collection

The medical records of all the patients with successful OncoPanel sequencing were reviewed to determine clinical and demographic characteristics, including sex, race (self-reported), ethnicity (self-reported), pathologic diagnosis, age at diagnosis, and disease stage at diagnosis. Characteristics for the specimen that underwent sequencing were also extracted, including timing of sample acquisition including relationship to treatment, and site of the tumor (primary site vs. metastatic). The pathologic diagnosis was classified according to the International Classification of Diseases for Oncology, version 3.2 (ICD-O-3.2)[46] following an expert committee review of the pathology report (S.J.F., A.A., S.A., K.L.L., P.B., A.J.C., K.A.J.) for each sequenced tumor sample. The expert committee included pediatric oncologists and neuro-oncologists, pediatric pathologists with neuropathology, and sarcoma expertise. Diagnoses were classified as extracranial or intracranial solid tumors, and further sub-classified into disease groupings per Supplementary Data 2.

For selected osteosarcoma, Ewing sarcoma, Wilms tumor, and neuroblastoma patients included in the analytic population, we used the PRISSMM™ data model[21] and trained data abstractors to obtain additional clinical data from the EMR. The PRISSMM™ model developed for adult solid tumors was adapted to these four pediatric malignancies by modifying cancer-type specific fields and adding specific prognostically important biomarkers. In addition, we obtained baseline imaging data from scan reports to enable derivation of Toronto Stage[24,47]. A curation guide was prepared for trained individuals to follow when abstracting data from the EMR and quality control was performed by dual abstraction for approximately 12% of cases. Clinical data were collected in RedCap[22,23].

### Assessment of cohort generalizability

To determine the extent to which this cohort represents the larger pediatric solid and brain tumor patient population, data from the Cancer in North America (CiNA) North American Association of Central Cancer Registries (NAACCR) 1995–2018 and the NCI's Surveillance, Epidemiology and End Results (SEER) Registries), submitted December 2020) were analyzed[11]. Registries include: California, Connecticut, Florida, Georgia, Hawaii, Idaho, Illinois, Iowa, Kentucky, Louisiana, Massachusetts, New Jersey, New Mexico, New York, Ohio, Pennsylvania, Seattle (Puget Sound), Tennessee, Texas, Utah, and Wisconsin.

These 23 NCCR registries represent 66% of all U.S. children, adolescents, and young adults ages 0–39 based on 2018 U.S. Populations.

### Sequencing and data analysis

Tumors were sequenced using the targeted NGS OncoPanel platform as previously described[48–50]. Sequencing was performed at the Center for Advanced Molecular Diagnostics (CAMD), a CLIA-certified clinical laboratory in the Department of Pathology at Brigham and Women's Hospital in Boston, Massachusetts. OncoPanel is a validated, targeted NGS panel of up to 447 cancer genes for the detection of single-nucleotide variants (SNV), insertions and deletions, and copy number alterations (CNA), as well as selected intronic regions for up to 60 genes for the detection of structural variants (SV). Samples were sequenced with multiple versions of OncoPanel (versions 1, 2, and 3) as gene coverage has expanded over time (gene coverage for each version utilized is provided in Supplementary Data 3). The variant allele fraction (VAF) cutoff used for OncoPanel reporting was 5%. However, lower VAF variants (minimum of 2.5%) were also included if assessed by the reporting molecular pathologist to be present with high confidence. A molecular pathology report was returned to treating providers at the time of sequencing.

For analyses of genomic variants, variant call files generated at the time of reporting were utilized. Additional filtering was applied to the existing pipeline output removing mutations found in either the ClinVar (v. 07_04_2019 release)[51] or gnomAD v2.1[52] databases. Tumor mutational burden was calculated by dividing the total remaining number of SNVs or small insertions and deletions (indels) by the total panel size for each version. SNVs and indels were classified as oncogenic if they were labeled as "Oncogenic", "Likely Oncogenic", or "Predicted Oncogenic" per the Memorial Sloan Kettering Precision Oncology Knowledge Base v3.4 (OncoKB)[53]. In addition, limited in-house curation was performed (YL, HG, SJF). Specifically, the following variants were further assessed for oncogenicity: (1) loss-of-function (LoF) mutations in tumor suppressor genes (TSG); (2) SNVs and Indels in genes on actionable mutation lists classified as variants of uncertain significance (VUS) and; (3) all fusions involving genes on the actionable mutation lists. OncoPrints were created using the ComplexHeatmap (v. 2.4.3) package[54].

### Genomic associations with molecularly targeted therapy

Genomic alterations were analyzed and matched to the actionable mutation lists (aMOI) of three precision oncology medicine basket clinical trials investigating targeted therapy directed by tumor profiling: NCI-COG Pediatric MATCH Screening Trial (NCT03155620)[18], NCI-MATCH Screening Trial (NCT02465060)[19], and ASCO TAPUR Study, Version 3 (NCT02693535)[20]. Specific genomic alterations in the participant tumor samples were considered as matches using the following rules: (1) Precise match on either MATCH trial aMOI list (same CNA, SNV, Indel, or LoF mutation for tumor suppressors) taking into account resistance mutations; or (2) Same gene and variant type (eg. activating fusion, amplification or oncogenic SNV/Indel in an oncogene and LoF mutation or deletion in a TSG).

For patients with a tumor variant matching a basket trial treatment arm, the medical record was reviewed to determine whether the patient received a molecularly targeted therapy in the same drug class as the basket trial treatment arm. For patients who received molecularly targeted therapy, the mechanism of obtaining treatment (on a clinical trial, via single-patient research protocol, or prescribed off-study) was assessed.

### Statistics and reproducibility

No statistical method was used to predetermine sample size. No data were excluded from the analyses. The experiments were not randomized. The Investigators were not blinded to allocation during experiments and outcome assessment.

**Reporting summary**

Further information on research design is available in the Nature Portfolio Reporting Summary linked to this article.

## Data availability

The genomic and clinical datasets generated and analyzed in this study were submitted to the National Cancer Institute's Childhood Cancer Data Initiative (CCDI), and are available in the database of Genotypes and Phenotypes (dbGaP): Study Accession phs002677.v1.p1 [https://www.ncbi.nlm.nih.gov/projects/gap/cgi-bin/study.cgi?study_id=phs002677.v1.p1]. These data are available under restricted access due to individual privacy concerns, and requests are managed by NCI's Data Access Committee. There are no restrictions on how long data will be made available. Full details of data access are available on the dpGAP webpage, but all additional queries may be sent to NCIDAC@mail.nih.gov. The comprehensive PRISSMMTM clinical data were shared with the Massachusetts State Cancer Registry, which is making it accessible to the National Childhood Cancer Registry (NCCR) and the CCDI. Annotation databases included public resources such as OncoKb, ClinVar, and gnomAD databases. Source data are provided with this paper.

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

## Acknowledgements

The authors would like to thank the patients and families who participated in the Profile Cancer Reseach Study. The authors would like to acknowledge the Profile study at Dana-Farber/Brigham and Women's Cancer Center and Dana-Farber/Boston Children's Cancer and Blood Disorders Center for generating the sequencing data used in this project. The authors would also like to acknowledge the DFCI Oncology Data Retrieval System (OncDRS) for the aggregation, management, and delivery of the clinical and operational research data used in this project. Dana-Farber/Harvard Cancer Center is supported in part by an NCI Cancer Center Support Grant #NIH 5 P30 CA06516. The NCI's Childhood Cancer Data Initiative (CCDI) provided supplement funding to share data generated by this project with the research community under grant No. 3P30CA008748-54S3W.

## Author contributions

**Conception and design:** S.J.F., B.J.R., N.I.L., B.E.J., and K.A.J. **Financial support:** N.I.L., B.E.J., and K.A.J. **Administrative support:** A.W., N.I.L., and K.A.J. **Provision of study materials or patients:** A.A., P.B., S.S., E.A.M., N.B.C., S.N.C., K.D.W., K.L.L., R.L.S., B.J.R., S.S.B., J.M.G.A., N.I.L., M.L.M., L.E.M., B.E.J., A.J.C., and K.A.J. **Collection and assembly of data:** S.J.F., H.G., A.W., Y.Y.L., D.D., A.A., S.A., P.B., P.K., T.M., K.I.L., P.S., M.M., L.E.M., E.C., S.N.B., W.B.L., N.I.L., A.J.C., and K.A.J. **Data analysis and interpretation:** S.J.F., H.G., A.W., Y.Y.L, A.A., S.A., P.B., T.M., S.N.B., E.C., B.J.R., N.I.L., M.L.M., B.E.J., A.D.C., A.J.C., and K.A.J. **Manuscript writing:** All authors. **Final approval of manuscript:** All authors. **Accountable for all aspects of the work:** All authors

## Competing interests

The following represents disclosure information provided by the authors of this manuscript. Relationships may not relate to the subject matter of this manuscript. Richard L. Schilsky, Research support: ASCO receives research grants from the following companies to support the TAPUR trial: Astra-Zeneca, Bayer, Boehringer-Ingelheim, Bristol-Myers Squibb, Genentech, Lilly, Merck, Pfizer, Seagen, Taiho. I do not personally receive any remuneration from these companies. I serve on the Board of Directors of the following companies: Clarified Precision Medicine, Leap Therapeutics, I serve as a consultant to the following companies: Bryologyx, Cellworks, Flatiron Health, Syapse, Zephyr AI I chair a DSMB for the following company: Toray Pharmaceuticals, Bruce E. Johnson, Post Marketing Royalties for EGFR mutation testing from DFCI, Paid Consultant to Novartis, Checkpoint Therapeutics, Astra Zeneca, Daichi Sankyo, GSK, Hummingbird Diagnostics, Genentech, Bluedot Bio, G1 Therapeutics, Jazz Pharmaceuticals, Merus, Abdera, and Simcere Pharmaceutical, Unpaid Member of Steering Committee for Pfizer, Research Support from Cannon Medical Imaging, Andrew D. Cherniak, Research support from Bayer, Paid Consultant to BirdsEye Bio, Alanna J. Church, Honoraria: The Jackson Laboratory Consulting or Advisory Role: AlphaSights, The Jackson Laboratory, Bayer Travel, Accommodations, Expenses: Bayer, Katherine A. Janeway, Honoraria: Foundation Medicine, Takeda, Consulting or Advisory Role: Bayer, Ipsen, Travel, Accommodations, Expenses: Bayer, The remaining authors declare no competing interests.
