## [Peer Review File · Nature Communications]

Molecular Profiling of 888 Pediatric Tumors Informs Future Precision Trials and Data-Sharing Initiatives in Pediatric CancerREVIEWER COMMENTS

Reviewer #1 (Remarks to the Author): expertise in paediatric intracranial genomics

In this paper the authors examine the utility of tumor sequencing (oncopanel) in almost 900 pediatric patients with either CNS or extra cranial solid tumors.

There is a systematic assessment of targets identified in these patients that would meeting eligibility in one of 3 precision trials and this demonstrates ~1/3 of cases have targetable alteration.

This report adds to the literature (including papers cited in one of the tables) and provides additional information re:actionable target identification in pediatric cancers(mainly at diagnosis but also includes small % at relapse). Paper is well written and easy to follow.

Comments:

1. The authors mention the three studies that they use to define Actionable targets. A supplemental table would be helpful to show the reviewer which targets are included in each trial
2. What VAF was used as cutoff?? Were all of the included actionable variants clonal?
3. In the discussion the authors do not suggest the next steps or implications of this paper. In other words should all newly diagnosed patients be referred for panel or other next gen sequencing. Or do the authors believe that their results may provide some guidance as to which patients benefit most

Reviewer #2 (Remarks to the Author): expertise in basket trial design

The paper of Forrest et al. reported a large cohort of pediatric patients whom solid tumors were sequenced with OncoPanel, a targeted next-generation DNA sequencing panel.

In about 6 years, more than 1100 patients consented to be enrolled in the Profile Cancer Research Study and about one third of them had at least one actionable mutation of interest. In the end, only 41 patients were treated with molecularly targeted therapy.

The paper is well written and could represent a reference to address clinical trial design and precision oncology practice. Indeed, design of clinical trials utilizing tumor molecular data is quite uncommon in pediatric setting respect to adult age oncology.

Moreover, comprehensive genome profiling to identify actionable targets allows a new model of precision oncology for patients not responding to standard therapies.

As minor comment, authors could consider to implement the Consort diagram reported in figure 1, adding the number of aMOI patients and the number of patients who received targeted therapies

Reviewer #3 (Remarks to the Author): expert in mutation of interest trials

Summary of manuscript: Forrest et al. report the profiling on 888 Pediatric solid tumors using a common next generation DNA sequencing panel (OncoPanel). Within this group, 58% were extra-cranial solid tumors and 42% were CNS tumors. Notably, 55% of this cohort had one of ten common pediatric tumors, with the remainder having one of 85 distinct rare pediatric cancer diagnoses (fewer than 25 patients per histology).

There are two frequently cited pediatric pan-cancer sequencing studies. One of these studies included a high percentage of leukemia patients (Ma et al, reference 16 in this manuscript). The second study by Grobner et al. (reference 17 in this manuscript). included the ten common pediatric solid tumor diagnoses, but 75 of the 85 rare cancer diagnoses in the current manuscript were not included in the publication. In addition to the differences in the types of tumors analyzed, the previous studies included transcriptomic as well as copy number variation and mutation spectrum. In contrast, the current study utilized OncoPanel which is validated, targeted 352 NGS panel of up to 447 cancer genes for detection of single-nucleotide variants (SNV), 353 insertions, and deletions, and copy number alterations (CNA), as well as selected intronic 354 regions for up

to 60 genes for the detection of structural variants (SV).

A major focus of this manuscript was to determine the frequency of identifying molecularly targeted abnormalities in this population of pediatric patients with solid tumors. Overall, 33% of patients had at least 1 oncogenic genomic alteration that matched to 132 a targeted treatment arm of at least one of three precision oncology basket trial studies. The genes with that most frequently contained aMOIs were: BRAF (10%), NF1 (4%), CDKN2A (4%), PI3KCA (2.4%), NRAS/KRAS (2.1%) BRCA2 (1.5%), ALK (1.2%), and FGFR1 (1.2%). As might be expected, the frequency of targetable abnormalities varied with glial tumors having the highest match rate (>60%) while Ewing Sarcoma and neuroblastoma had low match rates of 7 and 12%, respectively. However, only 14% of patients received a matched molecularly targeted therapy and the outcomes for these patients were not reported.

Comments: This is a well written paper with a large cohort of pediatric patients that included many rare types of pediatric cancer. The sequencing data of overlapping histologic entities are congruent with previous reports, but the current manuscript included profiling of new entities. These results are notable for indicating the feasibility of such large scale efforts and the potential for using such analyses to guide therapy. As with other studies of this type in adult patients, only a minority of patients actually received molecularly targeted therapy. The authors speculate that this might be due to missing treatment data, such that some patients had good outcomes with standard therapy never had need of salvage or later-line molecularly targeted therapy. If the authors could obtain some of this data it could improve the impact of this study. The addition of any outcomes data from targeted therapy compared with standard therapy would also be useful. Despite these limitations, these data are an invaluable contribution to our understanding of genomic abnormalities in pediatric solid tumors, especially histology types outside the ten most common pediatric diagnoses.

RESPONSE TO REVIEWERS' COMMENTS

Reviewer #1 (Remarks to the Author): expertise in paediatric intracranial genomics

In this paper the authors examine the utility of tumor sequencing (oncopanel) in almost 900 pediatric patients with either CNS or extra cranial solid tumors.

There is a systematic assessment of targets identified in these patients that would meeting eligibility in one of 3 precision trials and this demonstrates ~1/3 of cases have targetable alteration.

This report adds to the literature (including papers cited in one of the tables) and provides additional information re:actionable target identification in pediatric cancers(mainly at diagnosis but also includes small % at relapse). Paper is well written and easy to follow.

Comments:

1. The authors mention the three studies that they use to define Actionable targets. A supplemental table would be helpful to show the reviewer which targets are included in each trial.

Author Response: Thank you for your comment. As suggested by Reviewed 1, we have added a supplemental table to show which gene targets are included in each of the three precision oncology trials.

2. What VAF was used as cutoff?? Were all of the included actionable variants clonal?

Author Response: Thank you for your comment. The VAF cutoff used for OncoPanel reporting was 5%, however lower VAF variants with a minimum of 2.5% were also reported and included in this analysis if the molecular pathologist reviewing the case determined that there was sufficient evidence that the variant was present. This information has been added to the methods section of the manuscript.

3. In the discussion the authors do not suggest the next steps or implications of this paper. In other words should all newly diagnosed patients be referred for panel or other next gen sequencing. Or do the authors believe that their results may provide some guidance as to which patients benefit most. Add a few sentences about implications, importance of sequencing, clinical trial.

Author Response: Thank you for your comment. We have added a paragraph to the Discussion section of the manuscript discussing next steps and implications of this paper and the importance of continued development of guidelines regarding the use of molecular profiling for pediatric cancers. In the interim, we believe continued efforts to molecularly profile pediatric CNS and extracranial solid tumors, particularly ultra-rare and advanced cancers, for diagnostic, prognostic and therapeutic purposes are critically important.

Reviewer #2 (Remarks to the Author): expertise in basket trial design

The paper of Forrest et al. reported a large cohort of pediatric patients whom solid tumors were sequenced with OncoPanel, a targeted next-generation DNA sequencing panel.

In about 6 years, more than 1100 patients consented to be enrolled in the Profile Cancer Research Study and about one third of them had at least one actionable mutation of interest. In the end, only 41 patients were treated with molecularly targeted therapy.

The paper is well written and could represent a reference to address clinical trial design and precision oncology practice. Indeed, design of clinical trials utilizing tumor molecular data is quite uncommon in pediatric setting respect to adult age oncology.

Moreover, comprehensive genome profiling to identify actionable targets allows a new model of precision oncology for patients not responding to standard therapies.

As minor comment, authors could consider to implement the Consort diagram reported in figure 1, adding the number of aMOI patients and the number of patients who received targeted therapies.

Author Response: Thank you for your comments. As suggested by Reviewer 2, we have updated Figure 1 to include the number of aMOI patients and the number of patients who received targeted therapies.

Reviewer #3 (Remarks to the Author): expert in mutation of interest trials

Summary of manuscript: Forrest et al. report the profiling on 888 Pediatric solid tumors using a common next generation DNA sequencing panel (OncoPanel). Within this group, 58% were extra-cranial solid tumors and 42% were CNS tumors. Notably, 55% of this cohort had one of ten common pediatric tumors, with the remainder having one of 85 distinct rare pediatric cancer diagnoses (fewer than 25 patients per histology).

There are two frequently cited pediatric pan-cancer sequencing studies. One of these studies included a high percentage of leukemia patients (Ma et al, reference 16 in this manuscript). The second study by Grobner et al. (reference 17 in this manuscript). included the ten common pediatric solid tumor diagnoses, but 75 of the 85 rare cancer diagnoses in the current manuscript were not included in the publication. In addition to the differences in the types of tumors analyzed, the previous studies included transcriptomic as well as copy number variation and mutation spectrum. In contrast, the current study utilized OncoPanel which is validated, targeted 352 NGS panel of up to 447 cancer genes for detection of single-nucleotide variants (SNV), 353 insertions, and deletions, and copy number alterations (CNA), as well as selected intronic 354 regions for up to 60 genes for the detection of structural variants (SV).

A major focus of this manuscript was to determine the frequency of identifying molecularly targeted abnormalities in this population of pediatric patients with solid tumors. Overall, 33% of patients had at least 1 oncogenic genomic alteration that matched to 132 a targeted treatment arm of at least one of three precision oncology basket trial studies. The genes with that most frequently contained aMOIs were: BRAF (10%), NF1 (4%), CDKN2A (4%), PI3KCA (2.4%), NRAS/KRAS (2.1%) BRCA2 (1.5%), ALK (1.2%), and FGFR1 (1.2%). As might be expected, the frequency of targetable abnormalities varied with glial tumors having the highest match rate (>60%) while Ewing Sarcoma and neuroblastoma had low match rates of 7 and 12%, respectively. However, only 14% of patients received a matched molecularly targeted therapy and the outcomes for these patients were not reported.

Comments: This is a well written paper with a large cohort of pediatric patients that included many rare types of pediatric cancer. The sequencing data of overlapping histologic entities are congruent with previous reports, but the current manuscript included profiling of new entities. These results are notable for indicating the feasibility of such large scale efforts and the potential for using such analyses to guide therapy. As with other studies of this type in adult patients, only a minority of patients actually received molecularly targeted therapy. The authors speculate that this might be due to missing treatment data, such that some patients had good outcomes with standard therapy never had need of salvage or later-line molecularly targeted therapy. If the authors could obtain some of this data it could improve the impact of this study. The addition of any outcomes data from targeted therapy compared with standard therapy would also be useful. Despite these limitations, these data are an invaluable contribution to our understanding of genomic abnormalities in pediatric solid tumors, especially histology types outside the ten most common pediatric diagnoses.

Author Response: Thank you for your comment. We agree with Reviewer 3 that outcome data for pediatric patients treated with molecularly targeted therapy is very important. Obtaining this outcome data as well as the reasons why patients did or did not receive molecularly targeted therapy was not within the scope of this project, and is the subject of a future study. We have included this limitation in the discussion and have emphasized the importance of this future work.

REVIEWERS' COMMENTS

Reviewer #1 (Remarks to the Author):

The authors have responded adequately to all comments

Reviewer #2 (Remarks to the Author):

The authors have responded satisfactorily to all the comments raised and the paper has improved

Reviewer #3 (Remarks to the Author):

The authors have definitely responded to the previous review comments. This is an important body of work for the pediatric oncology field.